# Effectiveness of Physiotherapy in the Treatment of Temporomandibular Joint Dysfunction and the Relationship with Cervical Spine

**DOI:** 10.3390/biomedicines10112962

**Published:** 2022-11-17

**Authors:** Maria Daniela Crăciun, Oana Geman, Florin Valentin Leuciuc, Iulian Ştefan Holubiac, Daniela Gheorghiţă, Florin Filip

**Affiliations:** 1Interdisciplinary Research Centre in Motricity Sciences and Human Health, Ştefan cel Mare University of Suceava, 720229 Suceava, Romania; 2Neuroaesthetics Laboratory, Ştefan cel Mare University of Suceava, 720229 Suceava, Romania; 3Dentist’s Office Omnis Dental, Ştefan cel Mare University of Suceava, 720229 Suceava, Romania; 4County Hospital of Suceava, Ştefan cel Mare University of Suceava, 720229 Suceava, Romania

**Keywords:** temporomandibular joint, cervical spine, physiotherapeutic treatment, dysfunction, pain, orofacial area, muscles, exercises

## Abstract

Temporomandibular dysfunctions are a heterogeneous group of conditions involving the temporomandibular joints (TMJs) and periarticular musculoskeletal structures. This study aimed to evaluate the effectiveness of a physiotherapy program for TMJ dysfunctions and the relationship with cervical spine. The study design was a non-randomized clinical trial with two parallel treatment groups: 33 subjects in the experimental group that underwent conservative drug treatment and physiotherapy treatment, and 31 subjects in the control group that underwent only conservative drug treatment. The participants were examined at baseline and re-examined after 3 months. In this study there was a higher incidence of female subjects. After 3 months of treatment of the TMJs and cervical spine, pain decreased in both groups (*p* = 0001). Muscle testing at the cervical spine and temporomandibular level showed a decrease in pain and muscles spasms. The average percentage values of the Neck Disability Index (NDI) and the Jaw Functional Limitation Scale 8 (JFLS 8) decreased significantly in both groups, but especially in the experimental group (*p* = 0.001). Physiotherapy treatments could maintain the functional state at the temporomandibular and cervical levels, thus contributing to increasing the quality of daily life.

## 1. Introduction

Temporomandibular dysfunctions (TMDs) are a heterogeneous group of conditions involving the temporomandibular joints (TMJs) and periarticular musculoskeletal structures. The temporomandibular morpho-functional complex registers a frequent pathology, which generates discomfort, disability and a negative effect on the quality of daily life. In TMJ dysfunction, the mandible is the central morphological element of the facial movements. Thus, it could start in isolation, but the mandibular dynamics could extend the involvement to the whole orofacial area [1]. In the world, over 450 million people had chronic facial pain, of which 6% were men and 10% were women. The incidence increases with age, especially after the age of 40 years [1,2].

Cervical spine disorders are also musculoskeletal disorders, that could cause significant disability in the general population [3,4].

According to literature, about 30% of men and 43% of women have had neck pain in their lifetime, and the intensity of the pain increases with age [5].

Stiesch-Scholz investigated the incidence of cervical spine dysfunctions at patients with temporomandibular disorders (TMDs). The results demonstrated a bigger restriction in cervical rotation, cervical flexion and extension, hypomobility at the level of the joint facets and suboccipital area, and muscular sensitivity at the cervical level, dorsal level, and shoulder area. In prolonged cervical flexion posture associated with stress, the mandibular condyle is pushed back against the meniscal tissue, causing inflammation, pain, and its progressive degeneration [6,7].

TMJs and afferent muscles are innervated by the trigeminal nerve. Therefore, the pain in TMD could be perceived as like a headache. The occipital area and the back of the neck are innervated by the spinal nerves C2–C7. Convergence exists between the cervical spinal nerves from the superior area and the trigeminal nucleus, what forms the complex cervical trigeminal. The pain resulting from TMJ dysfunction or cervical dysfunction may start as a peripheral phenomenon, but in time it may also appear in the central area [8,9,10].

The neuroanatomical and neurophysiological interconnection between the orofacial area and the cervical spine involves the masticatory system and the posture. This relationship has shown that cervical posture disorders cause functional changes at the orofacial level during mouth opening, chewing, and swallowing. Recognizing the relationship between TMJs and cervical spine and their pathology could help dentists and physical therapists to treat pain and dysfunctions at this level much more effectively [7].

Physiotherapy is a noninvasive method that includes manual therapy, exercises, and physical procedures, it is used in the therapy of TMD and cervical spine. Rehabilitation focused on TMD is an essential element of treatment leading to pain reduction and improvement of the functions of TMJ and cervical spine and increasing the quality of daily life. Physiotherapy is one of the treatments that could constitute the prevention of pain and degenerative changes in the musculoskeletal system [11,12].

The present study aimed to highlight the role of physiotherapy in the treatment of TMJs dysfunctions and the relationship with cervical spine, by applying a specific treatment for 3 months.

## 2. Materials and Methods

The study design was a non-randomized clinical trial with two parallel treatment groups. Prior to the start of the study, all subjects signed informed consent forms for clinical and functional evaluation and plans of physiotherapy treatments.

This study was carried out in agreement with the Research Ethics Commission of Stefan cel Mare University of Suceava (no. 42/05.10.2021) respecting the norms and regulations in force.

A prospective study was performed to identify a correct evaluation and treatment protocol, which investigated a series of subjects with degenerative joint diseases who presented symptoms of temporomandibular and cervical dysfunctions. A total of 64 subjects aged between 35 and 75 years were selected after careful analysis of their medical and treatment history files. They were divided into two groups, depending on their availability: 33 subjects were the experimental group who underwent conservative drug treatment and physiotherapy treatment, and 31 subjects constituted the control group who followed only conservative drug treatment (Figure 1). From the experimental group, two subjects were excluded, and from the control group, four subjects were excluded. Patients who did not follow the complete treatment or gave up during the treatment were not included in the study.

This study was addressed to the selected subjects following a questionnaire with inclusion and exclusion criteria Table 1.

Patients with degenerative joint diseases at the temporomandibular level (primary disease) and at the cervical level (secondary disease), in accordance with the diagnosis criteria for TMDs (DC/TMD), were selected for this study. Those diagnoses were also confirmed by ultrasonography, X-ray, computed tomography (CT) imaging, or cone beam CT.

The participants were examined at baseline (initial treatment, IT) and re-examined after 3 months.

The evaluation focused on Axis I (physical diagnosis) in accordance with DC/TMD for Clinical and Research Applications: pain intensity assessment (Visual Analog Scale, VAS), presence of joint noises (such as clicks, popping, and crackling), active range of motion (TMJ—opening and closing the mouth, laterality, protrusion, retrusion; cervical joint head and neck—flexion, extension, lateral flexion and rotation), presence of myalgia and spasm (masseter, temporalis, internal and external pterygoid, sternocleidomastoid, upper trapezius, splenius, and semispinalis muscles) [13,14,15].

Functional assessment of the cervical spine was conducted using the Neck Disability Index (NDI) and the Jaw Functional Limitation Scale 8 (JLFS 8) [13,16,17].

Discussions were held regarding the clinical situation of each patient at the beginning and end of each session, with treatment being modified in accordance with individual evolution.

The drug treatment was recommended by the specialist practician (dentist and/or rheumatologist) and targeted the pain and inflammation of the orofacial, temporomandibular and cervical, myalgia and periarticular spasm, and joint degeneration. Nonsteroidal anti-inflammatory drugs have beneficial effects in patients with temporomandibular and cervical degenerative diseases. Treatments with ibuprofen, diclofenac, meloxicam, piroxicam or naproxen (such as Advil, Ibrufen, Flamexin, and Vimovo), in different doses for at least 2 weeks is recommended depending on the clinical situation of each patient [18]. In other categories of TMDs, treatments with analgesics, corticosteroids, antidepressants, anticonvulsants, benzodiazepines and opioids are recommended [18,19].

Physiotherapy aims to decrease local and referral pain, restore mandibular and cervical movements, reduce myalgia and spasm in the face and neck, increase temporomandibular and cervical function, and maintain a correct posture of the head, neck, torso and scapular–humeral girdles [7,11,12,20]. Therapeutic exercises for masticatory muscles and/or of the cervical spine are used to increase strength, coordination, stability, motor control and endurance.

The treatment was applied to groups of patients and personalized in accordance with the assessment. It was applied after a detailed analysis of each case.

The physiotherapeutic treatment of the patients in this study followed the following methodology:Three sessions per week/first 2 weeks;Two sessions per week/next 2 weeks; andOne session per week/up to 3 months.

The duration of one session was between 60 and 90 min depending on the general and local relaxation of the patient.

The patients selected for this study were evaluated before physiotherapeutic treatments (initial evaluation) and after a period of 3 months of treatment (final evaluation).

The following techniques were applied to the following areas: upper jaw, mandible, cervical spine and soft parts of the face and neck.

Evaluation and treatment of muscle spasms and myalgia for the following muscles: masseter, temporalis, internal pterygoid, external pterygoid, sternocleidomastoid (SCM); upper trapezius, splenius and semispinalis, head and neck massage, detection and treatment of the trigger points, passive, passive–active and active stretching exercises [12,20,21];mandible manipulation techniques (extra and intraoral) to increase the range of motion at the TMDj by performing movements of descent, laterality, protrusion and retrotrusion: caudal traction movements, mid-lateral gliding movements and ventral gliding movements [22,23];techniques for manipulating the cervical spine to increase range of motion by performing flexion, extension, lateral flexion, and head and neck rotation [24];exercises to prevent joint noises: exercises to increase tonus of the suprahyoid muscles by pushing the mandible backward, thus preventing wide and uncoordinated opening; the exercise is performed under visual control, at first with the mouth closed and the teeth touching very lightly [25,26];proprioceptive techniques at the TMJ: isometric coordination exercises, with mouth closed, half-open, and open; exercises to correct joint and muscle asymmetries by manual control applied at the cranio–mandibular level, and exercises to open the mouth with the trunk inverted [21];techniques to correct deglutition: swallowing a small amount of water, swallowing and speaking while holding a semi-hard small object between the dental arches [27];diaphragmatic respiration: patients were instructed on how to breathe for this technique in supine position, breathing properly from the sitting position, and finally from orthostatic position alternating the position of the head and neck;corrective techniques for the head, neck and torso from supine position, prone position, sitting position and standing, permanently maintaining visual control of the gesture performed [27,28]. Exercises to stabilize the deep cervical flexors (along the head and neck) maintain the correct posture of the cervical spine, improving muscle control at the craniofacial level [7,25].

All patients in this study received a home exercise program that included corrective postures; self massage; toning exercises for stabilizing muscles of the scapula and deep neck flexors; exercises to prevent joint noises; stretching exercises for the upper masticatory, cervical, and thoracic muscles. The subjects were recommended to repeat the program two times a day, with the duration of each session being 30 mins.

### Statistical Analysis

G power software version 3.1.9.7 (Düsseldorf, Germany) was used the statistical power of the sample size, and the total number of participants was calculated to be 54, (effect size = 0.9, α-error probability = 0.05, power = 0.9). Thus, a total of 64 subjects were registered for the trials.

The subjects were divided into two groups by using Excel software version 2019 (Suceava, Romania). The first column indicated the names of the subjects, and the second column showed the formula “=RAND ()”. For each subject, a number was assigned as 0 or 1. Afterwards, the column where the numbers were assigned were selected, clicked, sorted, and filtered (from where “sort smallest to largest” was selected).

Data were systematized and centralized in an SPSS 24.0 database and processed using appropriate statistical functions. A 95% confidence interval was used in data presentation. Primary indicators (minimum, maximum, and frequency), mean value indicators (mean, median), and dispersion indicators (standard deviation, standard error, and confidence interval for the mean) were used for descriptive statistical analysis. Skewness test (−2 < *p* < 2) was used to validate the normality of the data distribution for the examined continuous variables. Qualitative significance tests, such as the Chi^2^ test were used for comparing the distributions of frequencies. Odds ratio (OR) and relative risk (RR) were used to measure the association between exposure and outcome. RR refers to the ratio between the incidence of a disease in the experimental group and the incidence of the same disease in the control; the higher the RR, the greater the association between the disease and the risk factor. If RR is 1 or close to 1, the risk of developing the disease is the same in the presence or absence of the risk factor; if RR < 1, a negative association of the disease exists with the risk factor (not a risk factor but a protective factor).

Independent sample *t*-test was used for comparing the means of any two normally distributed variables and and paired sample *t*-test was used for peer group study. Correlation coefficient “Pearson” (r) represents the correlation of 2 variables from the same group, the direct/indirect correlation being given by the sign of the coefficient.

## 3. Results

In both groups, a higher number of female subjects could be observed. The experimental group consisted of 23 female subjects (69.7% vs. 30.3%; *p* = 0.049), with a mean age of 58 ± 12 years (*p* = 0.539).

The level of pain in the region of the TMJ (assessed by VAS) recorded quite high values in both groups, the control group (4.19 ± 1.49) and in the experimental group (4.15 ± 1.21). The values dropped, significantly in the control group (4.19 vs. 1.74; *p* = 0.001) and especially in the experimental group (4.15 vs. 0.97; *p* = 0.001).

The VAS pain at the level of the cervical spine presented average values of 4.61 ± 1.38 in the control group, and 4.55 ± 1.28 in the experimental group. The pain subsided significantly in both groups, especially in the experimental group (4.55 vs. 1.21; *p* = 0.001), as shown in Figure 2.

In the experimental group, at the beginning about 75% of increased VAS values were at the temporomandibular and cervical levels (r = 0.748; *p* = 0.001). After 3 months, the correlation remained as significant (r = 0.700; *p* = 0.001), suggesting that VAS values decreased in both areas (Figure 3).

The ROC curve confirmed the following (Figure 4):Female gender is a good predictor of initiation treatment (AUC = 0.623; 95% CI: 0.484–0.761; *p* = 0.092); long time suffering of the temporomandibular area over 1.5 years is a good predictor to determine the treatment plans, with a sensitivity of 73% and a specificity of 45% (AUC = 0.612; 95% CI: 0.472–0.752; *p* = 0.124).

After 3 months of physiotherapeutic treatment, joint noise testing showed a significant decrease in the experimental group. The frequency of joint noises decreased significantly, especially in right and left laterality (*p* < 0.001), mouth opening/closing (*p* < 0.005), and protrusion–retrusion (*p* < 0.013, Table 2).

The muscle testing at the cervical area highlighted a decrease in pain and spasm in both groups, with much lower values in the experimental group.

After 3 months, the frequency of myalgia decreased significantly in the experimental group, especially in SCM, right upper trapezius, and right and left sides (*p* < 0.001). On the other tested areas, the frequency of pain decreased (*p* < 0.05) and the muscle spasm disappeared in all patients (Table 3).

Muscle testing of masseter, temporalis, and internal and external pterygoid, muscles showed a decrease in pain after treatment in all tested patients, with lower values in the experimental group (Table 4).

Thus, in the experimental group, the temporomandibular muscle testing made before of the initiation of the treatment and after 3 months of treatment highlighted decreasing significant pain. The muscle spasm also disappeared in all patients, regardless of the tested area, except the right/left external pterygoid, which was maintained in two out of 33 patients.

In both groups, especially in the experimental group, the joint testing of the cervical spine showed a significant increase in joint mobility (*p* = 0.001).

Testing the level of range of motion at the cervical spine area showed an increase in all parameters after treatment, with more obvious increase in the experimental group (Table 5).

Testing the level of range of motion at the temporomandibular area found an increase in all parameters after treatment, with more increase in the experimental group (Table 6).

In patients with neck disability, the average percentage values of the NDI decreased significantly in both groups (44.39 vs. 23.68; *p* = 0.001), especially in the experimental group (45.94 vs. 10.36; *p* = 0.001) (Table 7) (Figure 5).

In patients with jaw functional limitation, the average percentage values of JFLS 8 decreased significantly in both groups (32.98 vs. 18.87; *p* = 0.001), especially in the experimental group (32.65 vs. 7.35; *p* = 0.001) (Table 8) (Figure 5).

In the experimental group, at the beginning of the study, a significant correlation was found between NDI and JFLS 8 (r = +0.683; *p* = 0.001). After 3 months of treatment, over 87% of the higher NDI scores were associated with higher JFLS 8 scores (r = +0.872; *p* = 0.001) (Figure 6). 

## 4. Discussions

TMD occurs at any age, and it has repercussions on the whole body, especially influencing cervical function. Women are more prone to TMDs, and in the present work, over 65% were women. A 2008 study by Landi showed that the increased dysfunction in women is due to hormonal influences that exist after a certain age [29,30,31]. Thus, female gender is a good predictor of initiating physiotherapy treatment.

The duration of orofacial pain may influence the estimation of pain intensity on VAS, being overestimated when the duration is short or underestimated when the pain persists for a longer period [32]. TMJ pain is associated with pain in the joints of the cervical spine, which affects the perception of clinical signs and response to treatment.

Evidence showed that the craniomandibular region and upper cervical spine are related from anatomical, biomechanical, and neurophysiological standpoints [6,33].

Due to the convergence between the orofacial and cervical regions in the trigeminocervical nucleus [3], upper cervical pain is perceived in any orofacial region, innervated by the trigeminal nerve, and pain in any orofacial structure innervated by the trigeminal nerve perceived in the cervical regions innervated by upper cervical nerves [3,34]. The pain originated and maintained either in the orofacial region or in the cervical region is integrated in the trigeminal cervical nucleus and sent to the superior centers where it is then modulated by descending mechanisms. This phenomenon triggers changes in motor activity in the masticatory and cervical muscles. These changes may lead to the development of masticatory and cervical dysfunction, as seen in patients with TMD [35]. The results obtained in the present work are in accordance with the results presented by Wiesinger et al. [36] for TMD and spine pain. The authors revealed strong comorbidity between these two conditions, suggesting that they may share risk factors and influence each other.

Orofacial pain causes local and general functional changes by adopting analgesic positions and changing body position [37]. Therefore, physiotherapists working with patients with TMD need to be able to identify and treat these deficiencies earlier to reduce the vulnerability of the cervical spine, thus helping to improve the functioning of the craniocervical system.

TMD affects the masticatory and neck muscles. Myalgia in cervical muscles (sternocleidomastoid, upper trapezius, and splenius of the head and neck) decreased in both groups, with a significant decrease in the group that also underwent physiotherapy treatment. This finding could also be seen in orofacial muscles: masseter, temporalis, internal pterygoid and external pterygoid, thus restoring the physiological functionality of the mandible. Aspects related to the evolution of cervical and orofacial muscle pain after manual therapy and physiotherapy were highlighted in other studies [22].

TMD is often asymmetrical and it could negatively affect not only mastication, swallowing, and breathing but also the amplitude of movement in different segments of the spine (cervical area in the transverse plane, thoracic area in the sagittal plane, and lumbar area in the frontal plane), thus causing changes in the upper and lower limbs. 

Balancing TMJs through exercise corrects the dysfunctions and changes the position of the center of gravity, which has effects on the mobility of the spine and the stability of the limbs [38,39].

A direct relationship exists between the movements of TMJ and cervical spine and posture [5]. The functional relationships between the two regions need to be systematically evaluated.

The results are in accordance with those of Wänman A and Marklund S, who showed a significant improvement in pain and jaw function in patients treated by physiotherapy [40].

Physiotherapy favored the reduction in pain and orofacial muscle spasm, an increase in range of motion and local functionality [41].

The physiotherapeutic treatment targets to decrease the existing symptoms through general and local treatment methods on the basis of the type of the disorder and its stage.

Considerable progress has been made in recent years, with the rehabilitation of these patients being a major concern, leading to a significant decrease in morbidity.

This study highlighted the effectiveness of physiotherapy treatments applied to temporomandibular and cervical areas in the case of an existing condition in one of the two regions. The TMJs and cervical spine have interconnected relationships through neuroanatomical and neurophysiological structures. The presence of a disease in one of the two areas influences the mutual symptomatology. The effectiveness of physiotherapeutic treatments applied to both regions over a period of 3 months demonstrated a significant reduction in symptoms at the temporomandibular and cervical levels.

## 5. Conclusions

The results confirmed the coexistence of signs and symptoms of temporomandibular and cervical dysfunction, sensitive and hyperalgesic muscle points, and functional limitations.

Physiotherapy is an important treatment that aims to restore motor function, relieve musculoskeletal pain, and reduce muscle spasm and inflammation.

The subjects who participated in this research study moderate levels of orofacial and cervical disability.

The evolution of patients with temporomandibular and cervical involvement was influenced by applied physiotherapeutic techniques.

The treatment of patients with TMD involves extensive management, taking into account not only the treatment of the jaw, but also the treatment that involves the entire cranio–cervical–mandibular system and the spinal complex.

Physiotherapy maintains functional status and increases the quality of life.

## Figures and Tables

**Figure 1 biomedicines-10-02962-f001:**
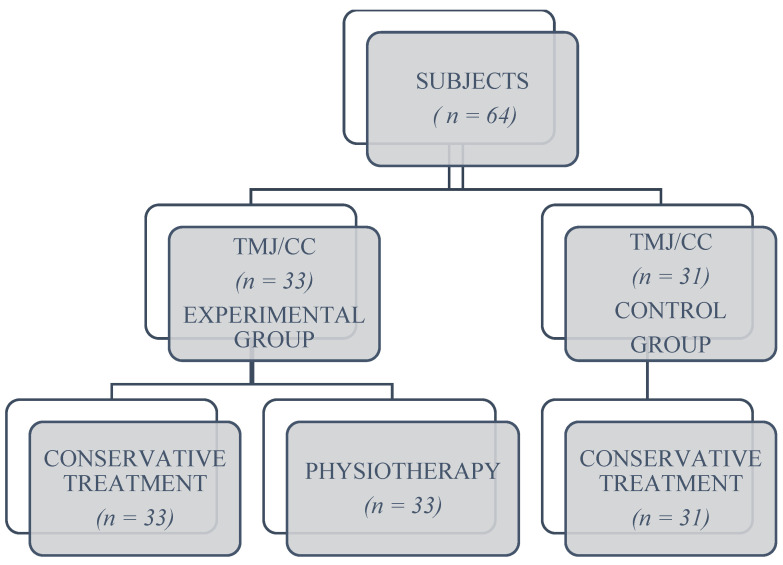
Group Design—Flowchart.

**Figure 2 biomedicines-10-02962-f002:**
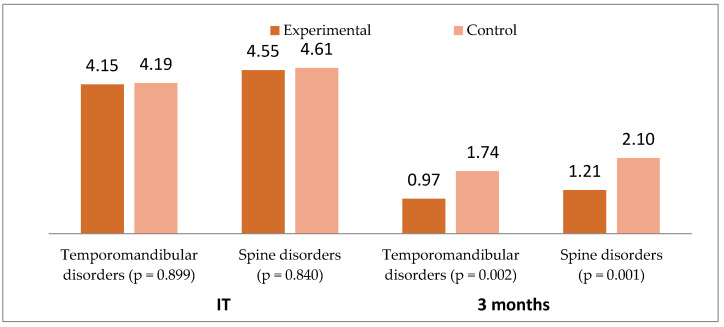
Comparative mean VAS values by study groups (initial (IT) vs. 3 months).

**Figure 3 biomedicines-10-02962-f003:**
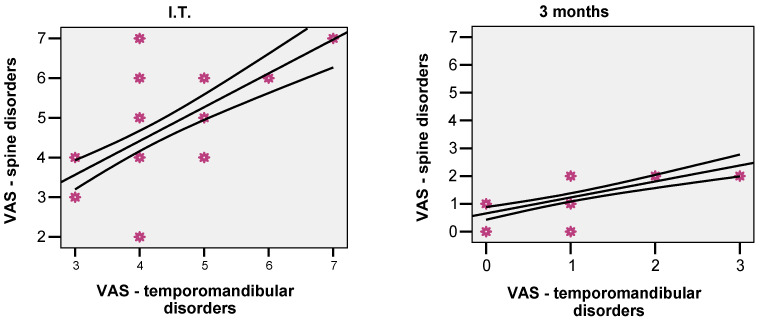
Correlation between VAS values in experimental group with temporomandibular and spine disorders (I.T.–initial vs. 3 months).

**Figure 4 biomedicines-10-02962-f004:**
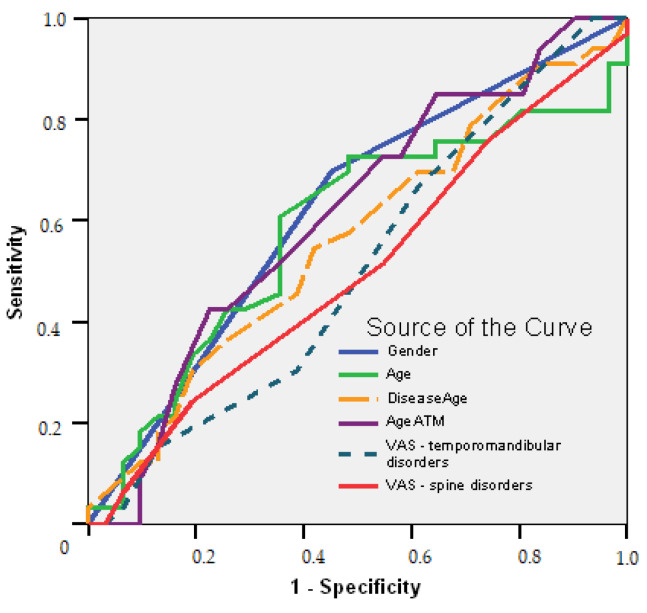
ROC curve. Predictive parameters in temporomandibular and cervical spine disorders.

**Figure 5 biomedicines-10-02962-f005:**
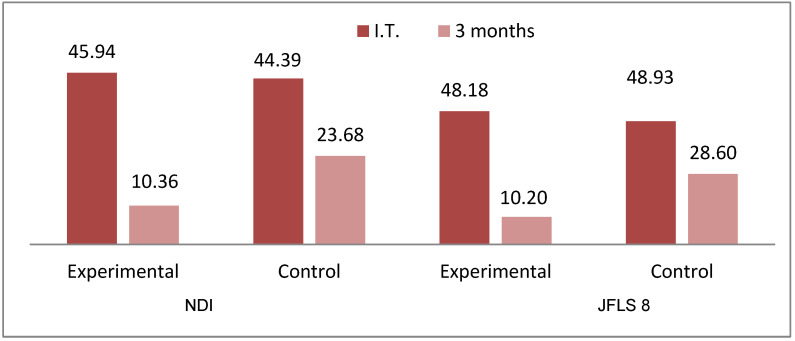
Comparative mean NDI and JLFS 8 by study group.

**Figure 6 biomedicines-10-02962-f006:**
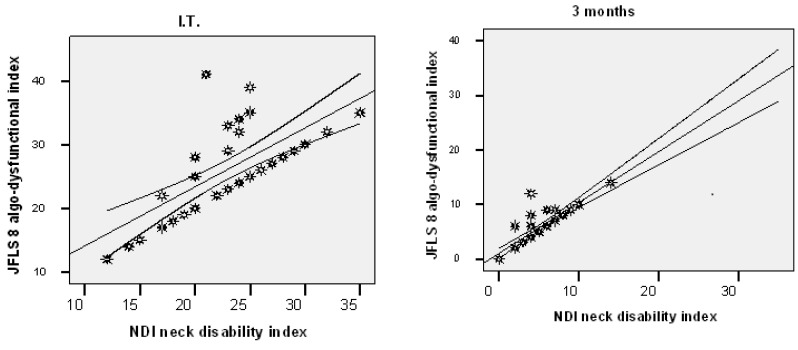
Correlation between NDI and JFLS 8 in the experimental group.

**Table 1 biomedicines-10-02962-t001:** Selection Criteria.

*SELECTION CRITERIA*	
*Inclusion Criteria*	*Exclusion Criteria*
neck pain	acute inflammation in the cervical area or temporomandibular level
pain in the temporomandibular region	feverish states
limiting the amplitude of movement at the cervical level	neoplasms at the level of the cephalic extremity or at other levels
limiting the amplitude of movement at the temporomandibular level	recent dental treatments- in the last 3 months
intra-articular noises in the last 30 days	recent head trauma—in the last 3 months
myalgia or spasms of the orofacial and cervical	drug specific treatment—in the last month
degenerative joint disease	physiotherapeutic specific treatment—in the last month
	mental disorders and mental illness
	neurological disorders
	ENT disorders
	systemic inflammatory conditions
	fibromyalgia

**Table 2 biomedicines-10-02962-t002:** Joint noise testing.

Parameters	Group		Tests Significance	
	Experimental	Control	*p* Value	95% CI
	(*n* = 33)	(*n* = 31)		
**Mouth opening/closing**				
Initial test *n* (%)	28 (84.8)	25 (80.6)	0.656 ^ns^	1.16 * (0.58–2.33)
After 3 months *n* (%)	11 (33.3)	21 (66.7)	**0.005** ^b^	0.24 * (0.08–0.68)
**Left–right laterality**				
Initial test *n* (%)	20 (60.6)	21 (67.7)	0.552 ^ns^	0.73 * (0.26–2.05)
After 3 months *n* (%)	5 (15.2)	17 (54.8)	**0.001** ^a^	0.15 * (0.05–0.48)
**Protrusion–Retrusion**				
Initial test *n* (%)	14 (42.4)	14 (45.2)	0.825 ^ns^	0.90 * (0.33–2.40)
After 3 months *n* (%)	4 (12.1)	12 (38.7)	**0.013** ^b^	0.22 * (0.06–0.78)

^a^ *p* < 0.001, ^b^ *p* < 0.05, ^ns^ no significance. * Relative risk for Chi-square test.

**Table 3 biomedicines-10-02962-t003:** Cervical spine muscle testing.

Parameters	Group		Tests Significance	
	Experimental (*n* = 33)	Control (*n* = 31)	*p* Value	95% CI
**SCM/Right**				
**Initial test**				
Pain *n* (%)	20 (60.6)	20 (64.5)	0.747	0.92 * (0.57–1.49)
Spasm *n* (%)	8 (24.2)	6 (19.4)	0.636	1.14 * (0.67–1.95)
**After 3 months**				
Pain *n* (%)	6 (18.2) ^a^	12 (38.7) ^b^	0.066	0.57 * (0.28–1.14)
Spasm *n* (%)	0 (0.0) ^b^	0 (0.0) ^b^	-	-
**SCM/Left**				
**Initial test**				
Pain *n* (%)	19 (57.6)	22 (71.0)	0.263	0.76 * (0.48–1.21)
Spasm *n* (%)	6 (18.2)	6 (19.4)	0.904	0.96 * (0.52–1.80)
**Aftee 3 months**				
Pain *n* (%)	6 (18.2) ^b^	11 (35.5) ^a^	0.116	0.61 * (0.31–1.22)
Spasm *n* (%)	0 (0.0) ^b^	0 (0.0) ^b^	-	-
**Right–Upper Trapezoid**				
**Initial test**				
Pain *n* (%)	23 (69.7)	23 (74.2)	0.689	0.90 * (0.54–1.49)
Spasm *n* (%)	2 (6.1)	5 (16.1)	0.192	0.53 * (0.16–1.74)
**After 3 months**				
Pain *n* (%)	7 (21.2) ^a^	13 (41.9) ^a^	0.072	0.59 * (0.31–1.13)
Spasm *n* (%)	0 (0.0) ^ns^	0 (0.0) ^b^	-	-
**Left–Upper Trapezoid**				
**Initial test**				
Pain *n* (%)	20 (60.6)	23 (74.2)	0.245	0.75 * (0.47–1.20)
Spasm *n* (%)	4 (12.1)	6 (19.4)	0.425	0.75 * (0.34–1.66)
**After 3 months**				
Pain *n* (%)	6 (18.2) ^a^	14 (45.2) ^a^	**0.019** ^b^	0.49 * (0.24–0.99)
Spasm *n* (%)	0 (0.0) ^b^	0 (0.0) ^b^	-	-
**Right–Splenius and Semispinal of the Head–Right**				
**Initial test**				
Pain *n* (%)	13 (39.4)	19 (61.3)	0.079	0.65 * (0.40–1.07)
Spasm *n* (%)	3 (9.1)	7 (22.6)	0.134	0.54 * (0.20–1.43)
**After 3 months**				
Pain *n* (%)	3 (9.1) ^b^	13 (41.9) ^ns^	**0.002** ^b^	0.30 *(0.11–0.85)
Spasm *n* (%)	0 (0.0) ^ns^	3 (9.7) ^ns^	**0.034** ^b^	-
**Left Splenius and Semispinal left head**				
**Initial test**				
Pain *n* (%)	13 (39.4)	19 (61.3)	0.079	0.65 *(0.40–1.07)
Spasm *n* (%)	3 (9.1)	8 (25.8)	0.073	0.48 *(0.18–1.30)
**After 3 months**				
Pain *n* (%)	3 (9.1) ^b^	14 (45.2) ^ns^	**0.001** ^b^	0.28 *(0.10–0.79)
Spasm *n* (%)	0 (0.0) ^ns^	2 (6.5) ^b^	0.085	-

^a^ *p* < 0.001, ^b^ *p* < 0.05, ^ns^ no significance. * Relative Risk for Chi-Square Test.

**Table 4 biomedicines-10-02962-t004:** Temporomandibular muscle testing.

Parameters	Group		Tests Significance	
	Experimental (*n* = 33)	Control (*n* = 31)	*p* Value	95% CI
**Masseter–Right**				
**Initial test**				
Pain *n* (%)	28 (84.8)	27 (87.1)	0.796	0.92 * (0.48–1.74)
Spasm *n* (%)	15 (45.5)	14 (45.2)	0.981	1.01 * (0.62–1.62)
**After 3 months**				
Pain *n* (%)	8 (24.2) ^a^	19 (61.3)	**0.002** ^b^	0.44 * (0.24–0.82)
Spasm *n* (%)	3 (9.1) ^a^	2 (6.5)	0.693	1.18 * (0.55–2.52)
**Masseter–Left**				
**Initial test**				
Pain *n* (%)	27 (81.8)	27 (87.1)	0.56	0.83 * (0.47–1.48)
Spasm *n* (%)	13 (39.4)	15 (48.4)	0.468	0.84 * (0.51–1.37)
**After 3 months**				
Pain *n* (%)	8 (24.2) ^a^	19 (61.3)	**0.002** ^b^	0.44 * (0.24–0.82)
Spasm *n* (%)	3 (9.1) ^a^	2 (6.5)	0.693	1.18 * (0.55–2.52)
**Temporalis–Right**				
**Initial test**				
Pain *n* (%)	21 (63.6)	26 (83.9)	0.064	0.63 * (0.41–0.99)
Spasm *n* (%)	7 (21.2)	5 (16.1)	0.602	1.17 * (0.67–2.02)
**After 3 months**				
Pain *n* (%)	8 (24.2) ^a^	16 (51.6)	**0.023** ^b^	0.53 * (0.29–0.99)
Spasm *n* (%)	0 (0.0) ^b^	0 (0.0)	-	-
**Temporalis–Left**				
**Initial test**				
Pain *n* (%)	21 (63.6)	25 (80.6)	0.127	0.69 * (0.44–1.08)
Spasm *n* (%)	7 (21.2)	5 (16.1)	0.602	1.17 * (0.67–2.02)
**After 3 months**				
Pain *n* (%)	9 (27.3) ^a^	16 (51.6)	**0.045** ^b^	0.59 * (0.33–1.04)
Spasm *n* (%)	0 (0.0) ^b^	1 (3.2)	0.226	-
**Right–Internal Pterygoid**				
**Initial test**				
Pain *n* (%)	15 (45.5)	21 (67.7)	0.071	0.65 * (0.40–1.04)
Spasm *n* (%)	7 (21.2)	9 (29.0)	0.47	0.81 * (0.44–1.49)
**After 3 months**				
Pain *n* (%)	4 (12.1) ^b^	14 (45.2)	**0.003** ^b^	0.35 * (0.14–0.86)
Spasm *n* (%)	0 (0.0) ^b^	1 (3.2)	0.226	-
**Left–Internal Pterygoid**				
**Initial test**				
Pain *n* (%)	14 (42.4)	21 (67.7)	**0.041** ^b^	0.61 * (0.38–0.99)
Spasm *n* (%)	7 (21.2)	9 (29.0)	0.47	0.81 * (0.44–1.49)
**After 3 months**				
Pain *n* (%)	4 (12.1) ^b^	14 (45.2)	**0.003** ^b^	0.35 * (0.14–0.86)
Spasm *n* (%)	0 (0.0) ^b^	1 (3.2)	0.226	-
**Right–External Pterygoid**				
**Initial test**				
Pain *n* (%)	31 (93.9)	30 (96.8)	0.588	0.76 * (0.33–1.76)
Spasm *n* (%)	27 (81.8)	16 (51.6)	**0.009** ^b^	2.20 * (1.08–4.49)
**After 3 months**				
Pain *n* (%)	10 (30.3) ^a^	17 (54.8)	**0.046** ^b^	0.60 * (0.34–1.04)
Spasm *n* (%)	2 (6.1) ^a^	4 (12.9)	0.345	0.62 * (0.20–1.98)
**Left–External Pterygoid**				
**Initial test**				
Pain *n* (%)	31 (93.9)	30 (96.8)	0.588	0.76 * (0.33–1.76)
Spasm *n* (%)	27 (81.8)	18 (58.1)	**0.036** ^b^	1.90 * (0.94–3.84)
**After 3 months**				
Pain *n* (%)	7 (21.2) ^a^	16 (51.6)	**0.011** ^b^	0.48 * (0.25–0.93)
Spasm *n* (%)	2 (6.1) ^a^	3 (9.7)	0.589	0.76 * (0.25–2.29)

^a^ *p* < 0.001, ^b^ *p* < 0.05, ^ns^ no significance. * Relative Risk for Chi-Square Test.

**Table 5 biomedicines-10-02962-t005:** Cervical spine joint testing.

Groups	Range of Motion at the Cervical Spine		
	Initial	3 Months	*p* Values for Paired Sample Test
**Flexion ^1^**			
Experimental (*n* = 33)	34.76 (±4.37)	42.67 (±2.00)	**0.001**
Control (*n* = 31)	33.97 (±3.42)	38.90 (±2.27)	**0.001**
*p* values for *t*-Student test Experimental vs. Control	*p* = 0.426	***p* = 0.001**	
**Extension ^1^**			
Experimental (*n* = 33)	31.36 (±4.55)	40.97 (±2.68)	**0.001**
Control (*n* = 31)	31.81 (±3.01)	36.32 (±2.89)	**0.001**
*p* values for *t*-Student test Experimental vs. Control	*p* = 0.650	***p* = 0.001**	
**Lateral flexion–Right ^1^**			
Experimental (*n* = 33)	35.15 (±4.57)	42.55 (±2.05)	**0.001**
Control (*n* = 31)	35.13 (±2.83)	39.65 (±1.85)	**0.001**
*p* values for *t*-Student test Experimental vs. Control	*p* = 0.981	***p* = 0.001**	
**Lateral flexion–Left ^1^**			
Experimental (*n* = 33)	35.21 (±4.33)	42.64 (±1.92)	**0.001**
Control (*n* = 31)	35.39 (±3.00)	39.77 (±1.93)	**0.001**
*p* values for *t*-Student test Experimental vs. Control	*p* = 0.852	***p* = 0.001**	
**Rotation–toward Right ^1^**			
Experimental (*n* = 33)	37.64 (±6.58)	53.30 (±6.62)	**0.001**
Control (*n* = 31)	36.74 (±3.44)	41.52 (±2.41)	**0.001**
*p* values for *t*-Student test Experimental vs. Control	*p* = 0.502	***p* = 0.001**	
**Rotation–toward Left ^1^**			
Experimental (*n* = 33)	37.70 (±6.52)	53.33 (±6.68)	**0.001**
Control (*n* = 31)	36.90 (±3.50)	41.45 (±2.46)	**0.001**
*p* values for *t*-Student test (experimental vs. control)	*p* = 0.550	***p* = 0.001**	

^1^ Unit measure grade (°).

**Table 6 biomedicines-10-02962-t006:** Temporomandibular joint testing.

Groups	Range of Motion at the Temporomandibular Joint		
	Initial	3 Months	*p* Values for Paired Sample Test
**Mouth opening/closing ^1^**			
Experimental (*n* = 33)	4.10 (±0.37)	4.80 (±0.17)	**0.001**
Control (*n* = 31)	4.11 (±0.30)	4.38 (±0.76)	**0.001**
*p* value for *t*-Student test Experimental vs. Control	*p* = 0.966	***p* = 0.003**	
**Right laterality ^1^**			
Experimental (*n* = 33)	0.84 (±0.09)	1.00 (±0.06)	**0.001**
Control (*n* = 31)	0.86 (±0.08)	0.92 (±0.07)	**0.001**
*p* values for *t*-Student test Experimental vs. Control	*p* = 0.320	***p* = 0.005**	
**Left laterality ^1^**			
Experimental (*n* = 33)	0.84 (±0.11)	1.00 (±0.06)	**0.001**
Control (*n* = 31)	0.86 (±0.10)	0.91 (±0.07)	**0.001**
*p* values for *t*-Student test Experimental vs. Control	*p* = 0.411	***p* = 0.001**	
**Protrusion ^1^**			
Experimental (*n* = 33)	0.48 (±0.05)	0.50 (±0.01)	**0.032**
Control (*n* = 31)	0.48 (±0.04)	0.49 (±0.03)	0.083
*p* values for *t*-Student test Experimental vs. Control	*p* = 0.915	*p* = 0.069	
**Retrusion ^1^**			
Experimental (*n* = 33)	0.37 (±0.05)	0.43 (±0.05)	**0.001**
Control (*n* = 31)	0.39 (±0.03)	0.41 (±0.03)	**0.012**
*p* values for *t*-Student test Experimental vs. Control	***p* = 0.015**	***p* = 0.022**	

^1^ Unit measure centimeter (cm).

**Table 7 biomedicines-10-02962-t007:** Neck Disability Index–NDI.

Groups	NDI		
	IT	T 3	*p* Values for Paired Sample Test
**Scores**			
Experimental (*n* = 33)	22.97 (±5.18)	5.18 (±3.29)	**0.001**
Control (*n* = 31)	22.19 (±5.95)	11.84 (±3.49)	**0.001**
*p* values for *t*-Student test Experimental vs. Control	*p* = 0.579	***p* = 0.001**	
**Percentage index**			
Experimental (*n* = 33)	45.94 (±10.35)	10.36 (±6.59)	**0.001**
Control (*n* = 31)	44.39 (±11.89)	23.68 (±6.99)	**0.001**

**Table 8 biomedicines-10-02962-t008:** Jaw Functional Limitation Scale 8–JFLS 8.

Groups	JFLS 8		
	IT	T 3	*p* Values for Paired Sample Test
**Scores**			
Experimental (*n* = 33)	26.12 (±7.02)	5.88 (±3.52)	**0.001**
Control (*n* = 31)	26.39 (±6.17)	15.10 (±3.72)	**0.001**
*p* values for *t*-Student test Experimental vs. Control	*p* = 0.873	***p* = 0.001**	
**Percentage index**			
Experimental (*n* = 33)	32.65 (±8.77)	7.35 (±4.41)	**0.001**
Control (*n* = 31)	32.98 (±7.72)	18.87 (±4.65)	**0.001**
*p* values for *t*-Student test Experimental vs. Control	*p* = 0.873	***p* = 0.001**	

## Data Availability

The data used to support the findings of this study are available from the corresponding author upon request.

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
