# Peer review of "Effectiveness of Physiotherapy in the Treatment of Temporomandibular Joint Dysfunction and the Relationship with Cervical Spine"

_biomedicines, 2022, doi:10.3390/biomedicines10112962_

Round 1
Reviewer 1 Report (Previous Reviewer 1)
Thank you for the improved manuscript. I would still recommend some more english editing regarding spelling mistakes and sentence structure.
Furthermore I still miss explanations under the table containing the units and what is shown.
Please check figure 3. The first part I.T seems to be swapped.
Please check figure 8. Positioning of the numbers.
Author Response
We would like to thank for the thorough reading of this manuscript, and also for the thoughtful comments and constructive suggestions, which help us to improve the quality of the manuscript. We have studied these comments carefully and have made corresponding corrections that we hope will meet your approval.
- I would still recommend some more english editing regarding spelling mistakes and sentence structure.
We send the manuscript to an certified editor, and waiting for the requested corrections.
- Furthermore I still miss explanations under the table containing the units and what is shown.
. We verified the tables and introduced under each table the explanations requested.
- Please check figure 3. The first part I.T seems to be swapped.
The figure 3 was not modified. It is the same figure. We corrected the translation (IT instead of TI) under the figure 3.
- Please check figure 8. Positioning of the numbers.
We modified the position of the numbers for figure 8.

Reviewer 2 Report (New Reviewer)
Dear authors,
Your paper seems of scientific interest and well structured, but some aspects have to be addressed.
In order to improve the visibility of Your work, I suggest to mention among the keywords some terms different from those used in the title.
In the abstract and in the method section, it is necessary to specify the specific study design. Is it a non randomized clinical trial? Please explain it.
Please provide the Institutional Review Board approval for this study also in the method section. In the same section, please report that all the participants signed an informed consent.
Please specify why, in Your opinion, there were no drop outs.
How the sample size was preliminary calculated? Was it a convenience one? Please specify it, according to similar studies available in scientific literature, such as the following:
- Farì G, Santagati D, Pignatelli G, Scacco V, Renna D, Cascarano G, Vendola F, Bianchi FP, Fiore P, Ranieri M, Megna M. Collagen peptides, in association with vitamin C, sodium hyaluronate, manganese and copper, as part of the rehabilitation project in the treatment of chronic low back pain. Endocr Metab Immune Disord Drug Targets [Internet]. 2022;22(1):108-15.
- Notarnicola A, Maccagnano G, Farì G, Bianchi F, Moretti L, Covelli I, Ribatti P, Mennuni C, Tafuri S, Pesce V, Moretti B. Extracorporeal shockwave therapy for plantar fasciitis and gastrocnemius muscle: Effectiveness of a combined treatment. J Biol Regul Homeostatic Agents [Internet]. 2020;34(1):285-90.
Please specify if, among the inclusion/exclusion criteria, there were also some treatments (i.e. drugs or therapeutic exercises) in the previous weeks or months.
Please check the text line spcing in the method section, it is inconsistent.
The discussion section well deepens the relationship between cervical and temporomandibular dysfunctions, while it could better deepen the effectiveness of physiotheraphy for treating temporomandibular disorders. To do that, I suggest the following and recent references:
- Ferrillo M, Nucci L, Giudice A, Calafiore D, Marotta N, Minervini G, d’Apuzzo F, Ammendolia A, Perillo L, de Sire A. Efficacy of conservative approaches on pain relief in patients with temporomandibular joint disorders: A systematic review with network meta-analysis. Cranio J Craniomandibular Sleep Prac [Internet]. 2022
Best regards
Author Response
Response to Reviewer 2 Comments
We would like to thank for the thorough reading of this manuscript, and also for the thoughtful comments and constructive suggestions, which help us to improve the quality of the manuscript. We have studied these comments carefully and have made corresponding corrections that we hope will meet your approval.
- In order to improve the visibility of Your work, I suggest to mention among the keywords some terms different from those used in the title.
.We mentioned more words as you requested in the keywords.
- .In the abstract and in the method section, it is necessary to specify the specific study design. Is it a non randomized clinical trial? Please explain it.
We made the requested corrections and explained in the manuscript.
- Please provide the Institutional Review Board approval for this study also in the method section. In the same section, please report that all the participants signed an informed consent.
We provided the request regarding the aproval of the Institutional Rewiew Board for the study, and mentioned your second request in the same section about informed consent forms.
- Please specify why, in Your opinion, there were no drop outs.
We made the requested corrections.
- How the sample size was preliminary calculated? Was it a convenience one? Please specify it, according to similar studies available in scientific literature.
We explained and specified what you ask for this request in the manuscript.
- Please specify if, among the inclusion/exclusion criteria, there were also some treatments (i.e. drugs or therapeutic exercises) in the previous weeks or months.
We specified in exclusion criteria.
- Please check the text line spacing in the method section, it is inconsistent.
We verified and made the needed corrections for the text line in the method section.
- The discussion section well deepens the relationship between cervical and temporomandibular dysfunctions, while it could better deepen the effectiveness of physiotheraphy for treating temporomandibular disorders.
- We verified and corrected the highlighted aspects.
- To do that, I suggest the following and recent references.
We introduced two recent references.
Best regards

This manuscript is a resubmission of an earlier submission. The following is a list of the peer review reports and author responses from that submission.
Round 1
Reviewer 1 Report
I would recommend to reject the paper.
Author Response
We would like to thank the editor and reviewers for the thorough reading of this manuscript, and also for the thoughtful comments and constructive suggestions, which help us to improve the quality of the manuscript. We have studied these comments carefully and have made corresponding corrections that we hope will meet your approval.
Please see the attachment

Reviewer 2 Report
In this manuscript, the research question is unclear to me. However, I find it admirable that the inclusion of patients was rigorously performed by the most recent standards as well as with medical imaging. Since the research question is not clearly stated, it is unclear if the prevalence of cervical spine complaints associated with TMD will be evaluated or that rather the effect of a 3 month intervention of physiotherapy on TMD will be studied. Depending on the research question, the statistical analysis should be altered appropriately.
Additionally, the English language needs extensive editing.
Title:
The title is not representative for the study
Introduction
The aim of the study is not clearly stated in the introduction. It is unclear to me what the purpose or target population is. For example:
In the introduction is clearly stated from link 81-86 that “The temporomandibular joints, muscles, ligaments, fascial connections, innervation and circulatory system are tight connected. Any dysfunction, occlusal condition, postural disorders, trauma of the superior part of the body may lead to a problem of the connected or adjacent components. Thus, an evaluation of the cervical spine it must be performed”
So what is the need fort his study? What news will it add and what is the aim of the study?
Line 101: it is peculiar that the authors state that physiotherapy of the cervical spina and TMJ may resolve headaches. What is the target population of the study? Patients with headache? Neck pain? TMD?
It would be better to add a reference stating that physiotherapy improves TMJ dysfunctions
Materials and methods
I have some questions about the palpation of the longus colli. To my opinion, this muscle is not palpable. In a publication of Jull et al 2008 (https://pubmed.ncbi.nlm.nih.gov/18804003/) the muscle activity was measured by a nasopharyngeal electrode.
The outcome parameters are not clearly descirbed
The intervention might differ extensively between participants. Could this have influenced the findings? Since motor control exercises, swallowin, mobilizations ? manipulations ? were performed on both the cervical spine and TMJ
Since the aim of the study is not clearly stated, it is hard to tell if the statistical analysis is appropriate. I would assume that when you perform a treatment during an intervention of 3 months, one would like to assess the effect of the treatment? Now odds ratios are measured, which cannot detect the effectiveness of the intervention. Please us a different statistical analysis
Minor:
P2. Line 47: please insert a reference for this statement
Author Response
We would like to thank the editor and reviewers for the thorough reading of this manuscript, and also for the thoughtful comments and constructive suggestions, which help us to improve the quality of the manuscript. We have studied these comments carefully and have made corresponding corrections that we hope will meet your approval.
Please see the attachment.

Reviewer 3 Report
The paper entitled “The Relationship between the Temporomandibular Joints and Cervical Spine—Their Ailments and the Concept of Physiotherapeutic Treatment” need a major revision before it can been considered for publication.
Abstract should report the aim of the study and conclusions should also be rewritten according to the results obtained by the authors.
English should be reviewed by a native speaker: for example check line 99….
Methods section should exclusively describe the study design.
Please correct the paper reporting and than resubmit.
Author Response

(The authors gave the same response as above.)
